# Exploring coagulation parameters as predictive biomarkers of Plasmodium infection: A comprehensive analysis of coagulation parameters

Zelalem Tesfaye[1], Adane Derso[2], Ayalew Jejaw Zeleke[2], Ayenew Addisu[2], Berhanu Woldu[3], Teshiwal Deress[4], Gebeyaw Getnet Mekonnen[2], Yalewayker Tegegne[2]*

1 Bichina Primary Hospital, Amhara Regional State Health Bureau, Bahir Dar, Ethiopia, 2 Department of Medical Parasitology, School of Biomedical and Laboratory Sciences, College of Medicine and Health Sciences, University of Gondar, Gondar, Ethiopia, 3 Department of Haematology and Immunohaematology, School of Biomedical and Laboratory Sciences, College of Medicine and Health Sciences, University of Gondar, Gondar, Ethiopia, 4 Department of Quality Assurance and Laboratory Management, School of Biomedical and Laboratory Sciences, College of Medicine and Health Sciences, University of Gondar, Gondar, Ethiopia

* tyalewayker@yhaoo.com

**Data Availability Statement:** All relevant data are within the manuscript and its Supporting Information files.

## Abstract

### Background

Malaria affects the intravascular environment, leading to abnormal coagulation activation, prolonged prothrombin time, and activated partial thromboplastin time. Despite the high prevalence of malaria in the study area, there has been little published research on the effects of Plasmodium infection on coagulation parameters.

### Objective

The aim was to assess the effect of malaria on basic coagulation parameters among patients attending Dembia Primary Hospital and Makisegnit Health Center.

### Methods

A cross-sectional study was carried out from January to March 2020. The study involved 120 participants. Blood specimens were collected, which were analyzed using a Huma Clot Due Plus analyzer. The collected data were entered into EpiData and exported to SPSS version 21 for analysis. Non-parametric statistical methods were employed to analyze the data. The results were considered statistically significant if the $p$-value was less than 0.05.

### Results

Individuals infected with Plasmodium exhibit coagulation disorders with elevated levels of PT (Prothrombin Time), APTT (Activated Partial Thromboplastin Time), and INR (International Normalization Ratio) in comparison to healthy controls. The median PT, APTT, and

**Funding:** The author(s) received no specific funding for this work.

**Competing interests:** The authors have declared that no competing interests exist.

**Abbreviations:** APTT, Activated Partial Thromboplastin Time; DIC, Disseminated Intravascular Coagulation; HBsAg, Hepatitis B Surface Antigen; HCV, Hepatitis C Virus; INR, International Normalization Ratio; PT, Prothrombin Time; RBC, Red Blood Cells; vWF, Von Willebrand Factor; WHO, World Health Organization.

INR values for infected cases were measured at 20.5 [8.6], 39.5 [17.9], and 1.8 [0.9], respectively, while healthy controls had measurements of 15.1 [2.5], 28.8 [8.3], and 1.3 [0.2] ($p \leq 0.001$). The severity of coagulation disorders increased with an increase in parasitemia levels. The type of Plasmodium species present had a significant impact on PT and INR values ($p \leq 0.001$), whereas APTT did not show any significant impact across the Plasmodium species ($p > 0.05$).

## Conclusion

The results of this study found that malaria has a substantial impact on various blood clotting parameters, including PT, APTT, and INR. Parasitemia severity is significantly associated with extended PT and INR, implying that the higher the parasitemia, the longer it takes for blood to clot. Furthermore, the study discovered that the PT and INR levels differed based on the type of Plasmodium species responsible for the infection.

## Introduction

Malaria poses a major global health concern, particularly in tropical and subtropical areas [1]. The disease can lead to serious complications such as cerebral malaria, fever with hepatopathy, jaundice, bleeding problems, and multi-organ dysfunction. These complications disrupt the body's normal processes and are associated with elevated fibrinogen turnover [2].

Malaria pathogenesis is related to the disturbance of the normal intravascular environment, which affects the coagulation system in different ways. This includes the activation of endothelial cells due to pro-inflammatory cytokines, the presence of circulating micro-particles and activated platelets, endothelial damage, and the interaction between parasite-derived proteins on the surface of infected red blood cells and coagulation receptors [3–6].

According to the World Health Organization (WHO) 2023 report, malaria caused an estimated 249 million cases, with more than 94% occurring in the WHO African region. The report also stated that there were 608,000 deaths reported globally in 2022, with around 580,000 of those occurring in Africa [7]. In Ethiopia, malaria is a significant public health problem, with nearly 70% of the population at risk of contracting the disease [8]. The predominant malaria-causing species are *P. falciparum* and *P. vivax*, accounting for 69% and 31%, respectively [9, 10]. Despite efforts to control the disease, Ethiopia has experienced an increase in confirmed malaria cases, with a 150% and 120% rise in 2023 compared to the same periods in 2021 and 2022 [11].

The coagulation factors work collaboratively to prevent excessive blood loss when injuries occur [12]. The coagulation system is increasingly recognized to play an important role in malaria, and several studies have shown altered levels of coagulation parameters in malaria patients [4, 13, 14]. A high parasitemia load, which causes hepatic microcirculation occlusion, results in abnormalities in the synthesis and secretion of coagulation factors and their inhibitors. Additionally, micro-particle formation from red blood cell macrophages contributes to the widespread activation of blood coagulation [15, 16]. Moreover, prolonged coagulation time can be attributed to severe malaria, especially cerebral malaria, and alteration of coagulation markers have been used as prognostic markers [2, 13]. Besides, in a recent study, it was reported that the protein derived from the parasite, histidine-rich protein II (HRP-II), induced

an inhibitory effect on the anti-coagulant protein antithrombin, which acts physiologically as an inhibitor of thrombin [17].

Activation of the blood coagulation system occurs concurrently with thrombocytopenia in severe *P. falciparum* malaria, preceding disseminated intravascular coagulation and coagulation failure [15, 16]. The pathogenesis of severe malaria, particularly cerebral malaria, is closely intertwined with alterations in the blood coagulation system [15, 18]. In such cases, coagulation disorder is observed in a small percentage of cases and is associated with bleeding. A study conducted in Ghana found that coagulopathy is common in acute symptomatic *P. falciparum* malaria, and the degree of coagulation abnormality correlates with parasitemia [19].

Measuring coagulation profiles disrupted by Plasmodium infection provides a highly sensitive way to evaluate the disease's severity and activity. Infected patients often exhibit accelerated coagulation, enhanced fibrinogen turnover, antithrombin-III consumption, and increased concentrations of fibrinogen degradation products. Parasitized erythrocytes and released cytokines contribute to this pro-coagulant state [20, 21]. There is no published data on the impact of Plasmodium infection on blood coagulation parameters in Ethiopia. Thus, this study aims to fill this gap and provide insights into the coagulation status of malaria patients in Ethiopia. Therefore, this study was conducted to assess the impact of Plasmodium infection on coagulation parameters among Plasmodium-infected study participants and healthy controls.

## Material and methods

### Study area and period

A comparative cross-sectional study was conducted from January to March 2020 at two healthcare facilities: Dembia Primary Hospital and Makisegnit Health Center. Dembia Primary Hospital is situated in the Dembia district of the central Gondar Administrative Zone, approximately 729 km to the Northwest of Addis Ababa, Ethiopia's capital city. The district covers an area of 1270 km$^2$ and is characterized by a malarious environment. The population in this area is approximately 263,000 and the majority relies on subsistence farming [22].

Makisegnit Health Center, located 42 km away from Gondar town, sits at an altitude of 1918 meters above sea level. The climate in the area is classified as tropical according to the Köppen-Geiger system. The region area experiences a greater amount of rainfall during the summer than in the winter, with an average precipitation of 1059 mm. The average temperature in Makisegnit is 20.2°C [23, 24].

### Sample size determination and sampling technique

According to rules of thumb that have been recommended by van Voorhis and Morgan, 30 participants per group are required to detect real differences, which leads to about 80% power [25]. To increase the accuracy of the result, double samples of each were considered. Then, the study participants proposed by the rule of thumb have to be increased twofold. Thus, a total of 120 study participants (60 malaria-infected and 60 healthy controls) were enrolled in the study using a convenient sampling technique.

### Study population

The study population consisted of patients who visited the outpatient department of the selected health facilities during the study period. The inclusion criteria for the Plasmodium-infected group included positive results for Plasmodium parasites by microscopy examination, while for the healthy control group, inclusion criteria were negative results for malaria

parasites by microscopy. Exclusion criteria for both groups included a history of chronic diseases, such as diabetes, hypertension, or renal failure, previous history of malaria, and use of anti-malarial drugs within the past two weeks.

## Data collection procedure

**Socio-demographic data collection.** Socio-demographic data from the study participants were collected using an interviewer-administered structured questionnaire. Clinical information and the history of malaria-negative individuals who attended Dembia Primary Hospital and Makisegnit Health Center were assessed by a trained nurse, while socio-demographic characteristics and patient history were collected by a trained laboratory professional.

**Blood sample collection and laboratory analysis.** Malaria diagnosis was conducted using blood film microscopy. Both thin and thick films were prepared on a single slide. The thin film was treated with methanol, stained with Giemsa stain, and examined under a 100× oil immersion objective. Parasite detection and density were confirmed using a thick blood film, while species were identified using a thin blood film. A slide was considered negative only after examining 100 good fields, and if a parasite was found, an additional 100 fields were examined [26, 27].

A laboratory technologist collected 5 ml of venous blood for coagulation profile tests. Approximately 3 ml of the sample was then transferred into a test tube with sodium citrate anticoagulant and centrifuged to obtain platelet-poor plasma for PT, APTT, and INR measurements. The remaining 2 ml of blood was used for HBsAg and HCV testing in a separate tube. These tests were performed at the University of Gondar Comprehensive Specialized Hospital Laboratory.

The coagulation parameters were measured using standard methods [28, 29] and analyzed with the Humaclot Due Plus coagulation analyzer. Thromboplastin reagent and calcium chloride solution were mixed and left undisturbed for 30 minutes at room temperature. The mixture was then incubated at 37˚C for 10 minutes. Test platelet-poor plasma was added to the test cuvette and incubated at 37˚C for 2 minutes. Then, a pre-warmed PT reagent was added, and clot formation time was recorded. The APTT test involved mixing the reagent and calcium chloride, pre-warming at 37˚C for 10 minutes, adding the test plasma to a cuvette, adding the APTT reagent, and recording the clot formation time. The HBV and HCV tests used colloidal gold-enhanced immunoassays for antigen detection in the nitrocellulose membrane test regions.

## Data management and analysis

Data entry and analysis were performed using EpiData and SPSS version 21, respectively. The collected data involved a multifaceted approach to comprehensively assess the coagulation parameters and their potential variations among groups. Initially, the normality of the coagulation parameters was analyzed using the Kolmogorov-Smirnov normality test, which showed that continuous variables were not normally distributed among each group. Levene's test for equality of variances was then employed to rigorously evaluate whether the variances of coagulation parameters remained consistent across groups within the study population. Subsequently, the data were analyzed using the Kruskal-Wallis test, and multiple comparisons were assessed using the Mann-Whitney U test. The Spearman correlation test was used to assess the relationship between basic coagulation profiles. Results for continuous variables were presented using the interquartile range (IQR) for each group. Finally, the study findings were presented in the form of appropriate texts and tables. In all statistical analyses, a *p-value* of less than 0.05 was considered statistically significant.

## Quality control

The reliability of the study findings was assured by implementing quality control measures during the whole process of the laboratory work. All materials, equipment, and procedures were adequately controlled. Known negative and positive control samples were used to check the functionality of the analyzer used during the study. Giemsa stain quality was checked by using the known positive and negative samples for every batch of prepared working solution. All samples were analyzed correctly for the reliability and accuracy of the result. Among the total samples, 15% were randomly selected and re-analyzed at the end by an experienced laboratory technologist who was blind to the first analysis result.

## Ethics approval and consent to participate

The study was approved by the School of Biomedical and Laboratory Science Ethical Review Committee, University of Gondar. A letter of support was secured from the Zonal health department, the district health office, and the Kebele administration. The objective of the study research was clearly explained to the study participants and those willing to take part in the study were also informed to withdraw from the study if they wanted to do so without any restriction. Then informed written consent was taken from the study participants and for less than 18 years participants' assent was obtained from their parents or caretakers/guardians, All the information obtained from the study participants was coded to maintain confidentially. In addition, patients who tested positive for malaria, HBsAg, or HCV were linked to a nearby health center.

## Results

### General characteristics of study participants

In this study, a total of 120 individuals (60 cases and 60 controls) aged from 3 to 56 years were included from Dembia Primary Hospital and Makisegnit Health Center. The median and interquartile age ranges were 18±14.3 and 14.5±18 years, respectively. Among them, 60.8% were males and 80.8% were from rural areas. More than half (52.5%) of the study participants were attending primary school. Regarding their occupation: 42.5% were students (Table 1).

### Coagulation tests abnormalities among cases and control group

About 95%, 63.3%, and 98.3% of Plasmodium species infected study participants showed a prolonged time of PT, APTT, and INR, respectively. Whereas apparently healthy controls showed 50%, 13.3%, and 75% prolonged coagulation time of PT, APTT, and INR, respectively. The prolonged PT, APTT, and INR results of individuals have shown a statistically significant association with Plasmodium infection ($p \leq 0.001$) (Table 2).

### Comparison of coagulation profiles of study participants

A non-parametric test was used to show the association of coagulation tests between cases and controls. The median [IQR] of PT, APTT, and INR were 20.5 [8.6], 39.4 [17.9], and 1.8 [0.9], respectively in Plasmodium-infected study participants. In comparison, the median [IQR] values of PT, APTT, and INR in apparently health controls, were 15.1 [2.5], 28.8 [8.3], and 1.3 [0.2], respectively. In the Mann-Whitney U test, the median [IQR] of PT, APTT, and INR of malaria-infected study participants were significantly higher than control (p ≤ 0.001) (Table 3).

**Table 1. Socio-demographic and clinical characteristics of study participants at Dembia Primary Hospital and Makisegnit Health Center from January to March 2020 (n = 120).**

| Variables | Variable category | Healthy control n (%) | Malaria-infected n (%) | Total n (%) |
|---|---|---|---|---|
| Gender | Male | 37(61.7) | 36(60.0) | 73(60.8) |
| | Female | 23(38.3) | 24(40.0) | 47(39.2) |
| Age in years | ≤ 10 | 0 | 9 (7.5) | 9 (7.5) |
| | 11–15 | 11 (3.3) | 11 (9.2) | 15 (12.5) |
| | 16–20 | 19 (15.8) | 19 (15.8) | 38 (31.7) |
| | 21–25 | 12 (10.0) | 6 (5.0) | 18 (15.0) |
| | 26–30 | 21 (17.5) | 5 (4.2) | 26 (21.7) |
| | ≥ 31 | 4 (3.3) | 10 (8.3) | 14 (11.7) |
| Residence | Urban | 16(26.7) | 7(11.7) | 23(19.2) |
| | Rural | 44(73.3) | 53(88.3) | 97(80.8) |
| Religion | Orthodox | 56(93.3) | 60(100.0) | 116(96.7) |
| | Muslim | 4(6.7) | 0(0) | 4(3.3) |
| Educational level | No formal education | 21(35.0) | 20(33.3) | 41(34.2) |
| | Primary school | 30(50.0) | 33(55.0) | 63(52.5) |
| | Secondary school | 7(11.7) | 6(10.0) | 13(10.8) |
| | Diploma and above | 2(3.3) | 1(1.7) | 3(2.5) |
| Marital status | Single | 41(68.3) | 40(66.7) | 81(67.5) |
| | Married | 17(28.3) | 18(30.0) | 35(29.2) |
| | Divorced | 2(3.3) | 2(3.3) | 4(3.3) |
| Occupation | Student | 19(31.7) | 32(53.3) | 51(42.5) |
| | Government employee | 13(21.7) | 14(23.3) | 27(22.5) |
| | Farmer | 22(36.7) | 13(21.7) | 35(29.2) |
| | Other | 6(10.0) | 1(1.7) | 7(5.8) |

## Comparison of coagulation tests between different groups of malaria parasite density

This finding showed median [IQR] of PT, APTT, and INR varied between different groups of malaria parasite density. Similarly, in the Kruskal-Wallis analysis, the median [IQR] of PT, APTT, and INR showed statistically significant differences between different groups of malaria parasite density of infected participants ($p < 0.001$). The median of PT was 20.45 and increased to 95% of the total cases, 87.5% of patients with parasitemia level +1, and 100% for +2, +3, and +4 parasite density levels. Similarly, the median of APTT was 39.5, it was elevated in 63.3% of the total cases and increased in 37.5% of patients with parasitemia level +1, 70.8% for +2, 100% for +3, and +4 parasite density level. Furthermore, the median of INR was 1.8 and it was

**Table 2. Coagulation test abnormalities among case and control groups of study participants, at Dembia Primary Hospital and Makisegnit Health Center from January to March 2020.**

| Variables | Variable category | Controls n (%) | Cases n (%) | *p*-value |
|---|---|---|---|---|
| PT | Normal | 30 (50.0) | 3 (5.0) | ($p \leq 0.001$) |
| | Prolonged | 30 (50.0) | 57 (95.0) | |
| APTT | Normal | 52 (86.7) | 22 (36.7) | ($p \leq 0.001$) |
| | Prolonged | 8 (13.3) | 38 (63.3) | |
| INR | Normal | 15 (25.0) | 1 (1.7) | ($p \leq 0.001$) |
| | Prolonged | 45 (75.0) | 59 (98.3) | |

**Table 3. Comparison of coagulation profiles among cases and controls using Mann Whitney U test at Dembia Primary Hospital and Makisegnit Health Center from January to March 2020.**

| Variables | Controls median [IQR] | Cases median [IQR] | p. value |
|---|---|---|---|
| PT/seconds | 15.1[2.5] | 20.5[8.6] | ($p \leq 0.001$) |
| APTT/seconds | 28.8[8.3] | 39.5[17.9] | ($p \leq 0.001$) |
| INR | 1.3[0.2] | 1.8[0.9] | ($p \leq 0.001$) |

prolonged to 98.3% of the total cases and increased in 95.8% of patients with parasitemia levels +1 and, 100% for +2, +3 and +4 parasite density levels had elevated INR (Table 4).

## Comparison of basic coagulation parameters among Plasmodium species

The coagulation parameter data including PT, APTT, and INR were not normally distributed among Plasmodium-infected study participants. Hence, the non-parametric test was used to compare the median difference of study participants. There was a statistically significant difference in PT and INR values between patients with *P. falciparum*, *P. vivax*, and mixed Plasmodium infection (*P. falciparum* and *P. vivax*) ($p \leq 0.001$). However, there was no significant difference in APTT value between patients with *P. falciparum*, *P. vivax*, and mixed infection ($p > 0.05$). The median of PT was 20.5. It increased in 95% of the total cases and increased in 100% of patients with *P. falciparum* malaria, 94.7% of *P. vivax*, and 85.7% of study participants with mixed infection. Similarly, the median of APTT was 39.5 and it increased in 63.3% of the total cases and increased in 77.8% of patients with falciparum malaria, 52.6% of the *P. vivax*, and 50% of mixed malaria cases. Furthermore, the median of INR was 1.8 and it was elevated in 98.3% of the total cases and increased in 100% of patients with *P. falciparum* malaria; *P. vivax*, and 92.9% mixed malaria patients (Table 5).

## Discussion

Malaria is a significant public health threat worldwide. The disease triggers the activation of various blood clotting processes, which lead to platelet activation, disruption of the balance between coagulation and anticoagulation factors, and endothelial cell dysfunction [31].

**Table 4. Comparison of coagulation tests among different categories of Plasmodium-infected study participants using the Kruskal-Wallis test at Dembia Primary Hospital and Makisegnit Health Center from January to March 2020.**

| Coagulation test | Parasite density | Frequency | Median | Interquartile range | p-value |
|---|---|---|---|---|---|
| PT/seconds | + | 24 | 16.7 | 2.7 | ($p \leq 0.001$) |
| | ++ | 24 | 21.3 | 5.1 | |
| | +++ | 6 | 31.2 | 14.2 | |
| | ++++ | 6 | 30.7 | 4.6 | |
| APTT/seconds | + | 24 | 29.5 | 14.4 | ($p \leq 0.001$) |
| | ++ | 24 | 42.1 | 19.1 | |
| | +++ | 6 | 50.5 | 65.0 | |
| | ++++ | 6 | 44.2 | 15.8 | |
| INR | + | 24 | 1.5 | 0.3 | ($p \leq 0.001$) |
| | ++ | 24 | 1.9 | 0.5 | |
| | +++ | 6 | 2.9 | 1.5 | |
| | ++++ | 6 | 2.8 | 0.5 | |

Note:+: 1–10 parasites /100 field; ++: 11–100 parasites/100 field; +++: 1–10 parasites/field and ++++: > 10 parasites/field [30].

**Table 5. Comparison of coagulation parameters among study participants infected with different Plasmodium species using the Kruskal-Wallis test at Dembia Primary Hospital and Makisegnit Health Center from January to March 2020.**

| Coagulation tests | Plasmodium Species | Number (%) | Median ± IQR | *p*. value |
|---|---|---|---|---|
| PT/seconds | *P. falciparum* | 27(45.0) | 26.2± 9.4 | ($p \leq 0.001$) |
| | *P. vivax* | 19(31.7) | 18.4± 4.2 | |
| | Mixed infection | 14(23.3) | 18.7± 4.5 | |
| APTT/seconds | *P. falciparum* | 27(45.0) | 43.5± 15.7 | 0.092 |
| | *P. vivax* | 19(31.7) | 37.6± 16.9 | |
| | Mixed (*P. falciparum* and *P. vivax*) | 14(23.3) | 36.7± 28.5 | |
| INR | *P. falciparum* | 27(45.0) | 2.4± 1.0 | ($p \leq 0.001$) |
| | *P. vivax* | 19(31.7) | 1.6± 0.4 | |
| | Mixed infection | 14(23.3) | 1.6± 0.4 | |

Despite its endemicity in Ethiopia, no prior study has explored the changes in coagulation parameters in patients infected with Plasmodium. This study investigated the impact of Plasmodium infection on coagulation parameters in patients in Ethiopia. The results demonstrate that malaria infection is associated with significant alterations in coagulation tests, particularly PT, APTT, and INR, with a positive correlation between parasite density and the degree of abnormality.

The current study discovered Plasmodium infected patients had significantly higher medians in coagulation tests such as PT, APTT, and INR when compared to healthy controls ($p \leq 0.001$). These results align with prior studies conducted in Sudan [5, 32, 33] and India [20, 34], which reported similar observations. The consistency of these findings across various studies conducted in different settings underscores the generalizability of the link between Plasmodium infection and coagulation abnormalities. Several possible mechanisms could explain these coagulation disturbances. One possibility is the damage and dysfunction caused by the parasite's interaction with endothelial cells. Dysregulation of the coagulation system may also be attributed to defects in coagulation inhibitors like antithrombin, protein C, and protein S. Another mechanism that could underlie coagulation abnormalities is impaired fibrinolysis and its impairment can lead to excessive clot formation. Additionally, the expression of tissue factors, potentially induced by the parasite's interaction with host cells, may play a significant role in these observed coagulation disturbances [35, 36].

Evidence of endothelium activation in malaria infection was obtained from Malawi which identified it as a key pathophysiological characteristic of malaria infection. Certain molecules released by the parasite's interaction with the endothelial receptors could trigger an amplified production of adhesion molecules. These adhesion molecules facilitate the binding of infected red blood cells to the endothelium, leading to the sequestration of parasites and the activation of inflammatory responses [37].

The current study also indicated that PT was prolonged in 95% of the Plasmodium-infected study participants and 50% of the control groups ($p \leq 0.001$). This result was supported by similar findings from Sudan [5, 32, 33], Thailand [38, 39], and India [20, 34, 40, 41]. However, another study from Thailand reported no difference in PT between the infected and control groups [42]. This variation might be explained by the diverse mechanisms employed by the malaria parasite to disrupt the balance between procoagulant and anticoagulant factors. Additionally, differences in ion concentrations, which can influence the coagulation cascade, might also contribute to these discrepancies [43].

In the current study, APTT was increased in 63.3% of the Plasmodium-infected participants, while only 13.3% of the apparently healthy controls showed an increase. The difference

in APTT prolongation between the Plasmodium-infected and healthy control was highly significant (p ≤ 0.001). These findings are consistent with studies conducted in Sudan [5, 32, 33] and India [20, 34]. However, there was no change in the APTT of Plasmodium-infected study participants compared to healthy controls [42]. This might be due to the fact that hemostasis plays a key role in the pathogenesis of malaria. Additionally, Plasmodium infections may have a different role in individuals or the development of complications due to the reduction of various factors in the cascade that proceeds to a hypocoagulable condition, leading to hemorrhagic manifestations [43].

Likewise, in this study, INR was prolonged in 98.3% of the Plasmodium-infected study participants and elevated in a median of 1.8 [0.9] of the cases with malaria, whereas it was prolonged in only 75% of apparently healthy control study participants and elevated with a median of 1.3[0.2] of apparently healthy control study participants. The INR elevation also showed a statistically significant difference between Plasmodium-infected and healthy control study participants ($p \leq 0.001$). These findings are consistent with similar studies conducted in Sudan [5] and India [44], where INR was also significantly prolonged in Plasmodium-infected patients compared to healthy controls. However, studies conducted elsewhere showed that the INR of Plasmodium-infected study participants was not elevated compared with healthy controls [42].

This might be due to Plasmodium infection causing endothelial dysfunction, which is the system used for the synthesis of substances involved in coagulation and fibrinolysis, including von Will brand factor (vWF), tissue plasminogen activator, plasminogen activator inhibitor, and protein S. Those lead to activation and consumption of coagulation factors, which results in prolonged PT, APTT and INR [45–47]. On the other hand, hemostatic dysfunction might have a role in advanced infection and the development of complications due to the reduced ion of various factors in the cascade that proceed to hypo coagulable condition which further leads to hemorrhagic manifestations [43].

In this study, significant differences were observed across parasite density in terms of PT, APTT, and INR (p ≤ 0.001). Study participants with higher parasite density exhibited elevated PT, APTT, and INR, indicating that increased parasitemia levels were associated with a higher likelihood of alterations in PT, APTT, and INR. The study also found a positive correlation between malaria parasite density and coagulation abnormalities in PT, APTT, and INR, consistent with findings from studies conducted in Thailand [38] and Sudan [5].

Furthermore, the study revealed that the median PT levels were higher in cases with *P. falciparum* (median 26.2 ± 9.4), followed by mixed infections (median 18.7 ± 4.5) and *P. vivax* (median 18.4 ± 4.2). Similarly, INR was higher in cases with *P. falciparum* (median 2.4 ± 1), followed by mixed infections (median 1.6 ± 0.4) and *P. vivax* (median 1.6 ± 0.4). Additionally, APTT levels were higher in *P. falciparum*-infected participants (median 43.5 ± 15.7), as well as in those with *P. vivax* (median 37.6 ± 16.9) and mixed infections (median 36.7 ± 28.5). In this study, a significant difference was observed in the effect on PT and INR among malaria-infected study participants with respect to Plasmodium species (p ≤ 0.001). However, Plasmodium species did not have a significant effect on APTT (p > 0.05). These results are in line with a study in India, which demonstrated elevated PT and INR [48]. Similarly, other studies in India also reported elevated PT, APTT, and INR in all Plasmodium species [49].

## Conclusion

This study has revealed that Plasmodium infections can significantly impact the coagulation profile of PT, APTT, and INR. Plasmodium species-infected study participants exhibited substantially higher median values of these parameters compared to healthy controls.

Furthermore, the severity of parasitemia was directly linked to the prolonged time of PT, APTT, and INR. The study findings emphasize the critical need for thorough monitoring of coagulation parameters in malaria patients, particularly those with severe infections.

## Supporting information

**S1 File.**
(XLSX)

**S2 File.**
(SAV)

**S3 File.**
(DTA)

## Acknowledgments

The authors of this study would like to extend their sincere gratitude to Dembia Primary Hospital, Makisegnit Health Center, and the University of Gondar for their support and assistance in conducting this research. Additionally, the authors would like to express their appreciation to the data collectors and study participants for their valuable contributions and willingness to participate in the study. Their cooperation and involvement were essential in the successful completion of this research.

## Author Contributions

**Conceptualization:** Zelalem Tesfaye, Teshiwal Deress, Gebeyaw Getnet Mekonnen, Yalewayker Tegegne.

**Data curation:** Zelalem Tesfaye.

**Formal analysis:** Zelalem Tesfaye, Adane Derso, Ayalew Jejaw Zeleke, Ayenew Addisu, Berhanu Woldu, Teshiwal Deress, Gebeyaw Getnet Mekonnen, Yalewayker Tegegne.

**Supervision:** Yalewayker Tegegne.

**Writing – original draft:** Zelalem Tesfaye, Adane Derso, Ayalew Jejaw Zeleke, Ayenew Addisu, Berhanu Woldu, Teshiwal Deress, Gebeyaw Getnet Mekonnen, Yalewayker Tegegne.

**Writing – review & editing:** Zelalem Tesfaye, Berhanu Woldu, Teshiwal Deress, Yalewayker Tegegne.

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
