## [Decision Letter · Decision Letter 0]

14 Feb 2024

PONE-D-24-00531Exploring Coagulation Parameters as Predictive Biomarkers of Malaria Infection: A Comprehensive Analysis of Coagulation ParametersPLOS ONE

Dear Dr. Asrat,

Thank you for submitting your manuscript to PLOS ONE. After careful consideration, we feel that it has merit but does not fully meet PLOS ONE’s publication criteria as it currently stands. Therefore, we invite you to submit a revised version of the manuscript that addresses the points raised during the review process.

We look forward to receiving your revised manuscript.

Kind regards,

Yash Gupta, Ph.D.

Academic Editor

PLOS ONE

Journal Requirements:

2. We note that your Data Availability Statement is currently as follows: All relevant data are within the manuscript and its Supporting Information files

Additional Editor Commenta:

Authors need to proofread the manuscript thoroughly to alleviate language concerns of the reviewers.

The manuscript discussion section is also weak, with lack of attention to the results described and previous reports.

Looking at the importance of study and novel findings, there is a need for extensive revision and a strong rebuttal to the queries raised by the reviewers.

Reviewers' comments:

Reviewer's Responses to Questions

**Comments to the Author**

1. Is the manuscript technically sound, and do the data support the conclusions?

Reviewer #1: Yes

Reviewer #2: No

2. Has the statistical analysis been performed appropriately and rigorously? 

Reviewer #1: Yes

Reviewer #2: I Don't Know

3. Have the authors made all data underlying the findings in their manuscript fully available?

Reviewer #1: Yes

Reviewer #2: Yes

4. Is the manuscript presented in an intelligible fashion and written in standard English?

Reviewer #1: Yes

Reviewer #2: No

5. Review Comments to the Author

Reviewer #1: Exploring Coagulation Parameters as Predictive Biomarkers of Malaria Infection: A Comprehensive Analysis of Coagulation Parameters

The authors have shown that malaria profoundly influences coagulation parameters, Prothrombin Time (PT), Activated Partial Thromboplastin Time (APTT), and International Normalized Ratio (INR). Data analysis reveals that malaria-infected individuals show coagulation abnormalities. Notably, the severity of these coagulation disorders escalates with increasing parasitemia levels, suggesting a direct correlation between parasitemia severity and coagulation time. This research contributes to our understanding of malaria’s impact on coagulation parameters, the study also opens the prospects of additional research to understand the mechanisms of these effects and to explore potential treatment strategies. This study has been continued research in this area to enhance our understanding and management of malaria’s effects on coagulation.

Comments:

1) The manuscript is well written and very well-organized conveying its research outcome clearly.

2) All the references are relevant in context to the manuscript.

3) On Page 3 Background section, 4th paragraph, the lines

“A high parasitemia load causing hepatic microcirculation occlusion results in abnormalities in the synthesis ……………………………. the widespread activation of blood coagulation”

Authors should discuss the factors that would leads to prolongation of blood coagulation as indicated by the results.

4) Authors have collected samples from endemic region however, have the authors considered analyzing the factors like reinfection of the parasite or have they also considered any pre-medication, besides malaria drug since these may also affect the blood coagulation.

5) In the result section authors have considered wide range age. The normal range of the PT value changes with age and should be carefully analyzed since small changes may lead to different out-come. (reference-https://www.ncbi.nlm.nih.gov/pmc/articles/PMC4992153/)

Reviewer #2: 1. The English writing of the manuscript is not well. It should be written in an elegant way.

2. Reframe the sentence, “Despite the high prevalence of malaria in the study area, there has been little published research on the effect of malaria infection on coagulation parameters” as malaria is a disease not infection.

3. In the result section the sentence, “Individuals infected with malaria exhibit coagulation disorders with elevated levels of PT, APTT, and INR in comparison to healthy controls” is using the acronyms. Authors should use full forms first time after that the acronyms.

4. Manuscript is lacking the in depth knowledge of about the basic molecular mechanism of endothelial activation in malaria. Authors should work on the basic molecular pathways of endothelial activation. I will not recommend this manuscript for publication.

6. PLOS authors have the option to publish the peer review history of their article (what does this mean?). If published, this will include your full peer review and any attached files.

Reviewer #1: No

Reviewer #2: **Yes: **Mradul Mohan

---

## [Author Response · Author response to Decision Letter 0]

14 Mar 2024

Manuscript submission ID: PONE-D-24-00531

Exploring coagulation parameters as predictive biomarkers of plasmodium infection: a comprehensive analysis of coagulation parameters

PLOS ONE

Dear PLOSE ONE Academic Editor,

We appreciate the invaluable opportunity to enhance our manuscript based on the constructive comments and suggestions provided. We genuinely thank both you and the reviewers for your dedication in offering insightful feedback and acknowledging the considerable time and effort invested.

In our revised submission, we meticulously integrated the suggested ideas, made essential corrections, and provided clarifications to address the comments from both the academic editor and the two reviewers. Enclosed herewith are detailed responses from the authors, addressing each comment and suggestion provided by the academic editor and the reviewers. We believe these revisions contribute significantly to the overall improvement of the manuscript.

Authors’ response to the editorial comments 

Dear Editor,

Thank you for your relevant comments. 

We have made modifications to the manuscript as per the comments given as stated below.

Editorial comments: 

1. Please ensure that your manuscript meets PLOS ONE's style requirements, including those for file naming. The PLOS ONE style templates can be found at https://journals.plos.org/plosone/s/file?id=wjVg/PLOSOne_formatting_sample_main_body.pdfand
https://journals.plos.org/plosone/s/file?id=ba62/PLOSOne_formatting_sample_title_authors_affiliations.pdf

Authors’ response: We have thoroughly reviewed the manuscript format and file naming requirements specified by PLOS ONE, and have adjusted our documents accordingly. We have also corrected the authors' affiliation symbol usage according to the template. 

2. We note that your Data Availability Statement is currently as follows: All relevant data are within the manuscript and its Supporting Information files

Please confirm at this time whether or not your submission contains all the raw data required to replicate the results of your study. Authors must share the “minimal data set” for their submission. PLOS defines the minimal data set to consist of the data required to replicate all study findings reported in the article, as well as related metadata and methods (https://journals.plos.org/plosone/s/data-availability#loc-minimal-data-set-definition).

- The values behind the means, standard deviations, and other measures reported;

Authors’ response: We appreciate your diligence in ensuring data transparency and reproducibility. In response to your inquiry, we have uploaded the raw data, including the values for measures such as mean, standard deviation, and other relevant data points, as a supporting information file in the submission system. We believe that this dataset encompasses the "minimal data set" criteria outlined by PLOS, providing the necessary information to replicate the study findings reported in the article. 

3. PLOS requires an ORCID iD for the corresponding author in Editorial Manager on papers submitted after December 6th, 2016. Please ensure that you have an ORCID iD and that it is validated in Editorial Manager. To do this, go to ‘Update my Information’ (in the upper left-hand corner of the main menu), and click on the Fetch/Validate link next to the ORCID field. This will take you to the ORCID site and allow you to create a new iD or authenticate a pre-existing iD in Editorial Manager. Please see the following video for instructions on linking an ORCID iD to your Editorial Manager account:

https://www.youtube.com/watch?v=_xcclfuvtxQ

Authors’ response: The corresponding author has already obtained and validated the ORCID iD in Editorial Manager, following the step-by-step process you provided. We appreciate your guidance and confirm that the necessary actions have been taken to comply with PLOS requirements. 

4. Additional editorial comments:

Authors need to proofread the manuscript thoroughly to alleviate the language concerns of the reviewers.

The manuscript discussion section is also weak, with a lack of attention to the results described and previous reports.

Looking at the importance of the study and novel findings, there is a need for extensive revision and a strong rebuttal to the queries raised by the reviewers.

Authors’ response: We have thoroughly proofread the revised submission, addressing language concerns. Additionally, we have strengthened the discussion section by providing more attention to the results and incorporating relevant previous reports. We believe these revisions enhance the manuscript's quality in light of the study's importance and novel findings. We tried to prepare to provide a strong rebuttal to the queries raised by the reviewers. 

The following are comments given by reviewers’ and authors’ point-by-point responses (The reviewer's comments are numbered and the authors’ responses are underlined)

Reviewer # 1

1. The manuscript is well written and very well-organized conveying its research outcome clearly

Authors’ response: We appreciate your positive feedback on our manuscript, acknowledging its clarity and organization in conveying the research outcome. We have also meticulously worked on further refining and editing the write-up in the revised submission. 

2. All the references are relevant in context to the manuscript

Authors’ response: Thank you for your insight 

3. On Page 3 Background section, 4th paragraph, the lines

“A high parasitemia load causing hepatic microcirculation occlusion results in abnormalities in the synthesis ……………………………. the widespread activation of blood coagulation”

Authors should discuss the factors that would leads to prolongation of blood coagulation as indicated by the results

Authors’ response: We appreciate your insightful comment in this regard. In response to your query, we have expanded our discussion on the factors influencing the prolongation of blood coagulation time, as outlined below. Notably, prolonged coagulation time can be attributed to severe malaria, particularly cerebral malaria, where alterations in coagulation markers have been identified as valuable prognostic indicators. Additionally, a recent study has highlighted the inhibitory effect of histidine-rich protein II (HRP-II), a protein derived from the parasite, on the anti-coagulant protein antithrombin. Antithrombin physiologically acts as an inhibitor of thrombin, and this interaction adds another layer to our understanding of the complex dynamics influencing blood coagulation in the context of severe malaria. 

4. Authors have collected samples from endemic region however, have the authors considered analyzing the factors like reinfection of the parasite or have they also considered any pre-medication, besides malaria drug since these may also affect the blood coagulation.

Authors’ response: Regarding the collection of samples from an endemic region, we want to clarify that our exclusion criteria, as outlined in the manuscript, were designed to account for potential confounding factors such as parasite reinfection and pre-medication. Specifically, study participants with a history of anti-malaria drug use within the past two weeks were excluded. This precaution was taken because we hypothesized that such pre-medication might influence the coagulation profile. In addition to excluding participants with recent anti-malaria drug use, we also excluded individuals with a previous history of malaria. This decision aimed to eliminate the possibility of including patients with malaria re-infection, as their condition might introduce variables affecting blood coagulation.

5. In the result section authors have considered wide range age. The normal range of the PT value changes with age and should be carefully analyzed since small changes may lead to different out-come. (reference https://www.ncbi.nlm.nih.gov/pmc/articles/PMC4992153/)

Author response: We have revised the categorization of the age groups for our study participants by implementing five distinct age intervals. After careful consideration, we observed that the number of participants below 10 years old and those above 30 years old was relatively low. Consequently, to enhance the clarity and statistical robustness of our analysis, we decided to merge participants aged ten years and below into a single category labeled as “≤ 10 years.” Similarly, participants aged 31 years and above have been combined into a category denoted as “≥ 31 years.” This modification was made to ensure a more balanced distribution of participants across age intervals and to address the limited frequency of individuals in the extreme age ranges, thereby optimizing the reliability of our study outcomes.

Reviewer # 2

1. The English writing of the manuscript is not well. It should be written in an elegant way

Authors’ response: In the revised submission of the manuscript, we have done significant language editing. 

2. Reframe the sentence, “Despite the high prevalence of malaria in the study area, there has been little published research on the effect of malaria infection on coagulation parameters” as malaria is a disease not infection.

Authors’ response: We have replaced malaria with plasmodium, as malaria is a disease, not an infection. We did the same thing throughout the revised submission manuscript document.

3. In the result section the sentence, “Individuals infected with malaria exhibit coagulation disorders with elevated levels of PT, APTT, and INR in comparison to healthy controls” is using the acronyms. Authors should use full forms first time after that the acronyms.

Authors’ response: PT (Prothrombin Time), APTT (Activated Partial Thromboplastin Time), and INR (International Normalization Ratio), are the extended forms of the abbreviations that we have included in the abstract’s result section, as we should use the full forms when we used it for the first time. 

4. Manuscript is lacking the in-depth knowledge of about the basic molecular mechanism of endothelial activation in malaria. Authors should work on the basic molecular pathways of endothelial activation. I will not recommend this manuscript for publication.

Authors’ response: Thank you for taking your valuable time to review our manuscript. We greatly appreciate your feedback and the opportunity to address your concerns.

While we understand your point regarding the basic molecular mechanism of endothelial activation in malaria disease, we would like to emphasize that our manuscript focuses primarily on the analysis of coagulation parameters as predictive biomarkers of plasmodium infection. We aimed to provide a comprehensive analysis of these parameters and their potential significance in the context of malaria.

We acknowledge the importance of understanding the molecular pathways of endothelial activation in malaria, and we agree that further research in this area is warranted. However, we believe that our manuscript contributes valuable insights into coagulation parameters as a potential predictive biomarker for Plasmodium infections. By exploring these parameters comprehensively, we hope to shed light on their role in the diagnosis and prognosis of plasmodium infections.

We would be more than willing to include a section discussing the basic molecular pathways of endothelial activation, as you have suggested. This addition would enhance the overall understanding of the topic and provide a more comprehensive analysis for the readers. We believe that by incorporating this additional information, our manuscript would be suitable for publication.

---

## [Decision Letter · Decision Letter 1]

27 Mar 2024

Exploring coagulation parameters as predictive biomarkers of Plasmodium infection: a comprehensive analysis of coagulation parameters

PONE-D-24-00531R1

Dear Dr. Asrat,

We’re pleased to inform you that your manuscript has been judged scientifically suitable for publication and will be formally accepted for publication once it meets all outstanding technical requirements.

Kind regards,

Yash Gupta, Ph.D.

Academic Editor

PLOS ONE

Additional Editor Comments (optional):

The revised manuscript is fit for the publication in PlosOne

Reviewers' comments:

Reviewer's Responses to Questions

**Comments to the Author**

1. If the authors have adequately addressed your comments raised in a previous round of review and you feel that this manuscript is now acceptable for publication, you may indicate that here to bypass the “Comments to the Author” section, enter your conflict of interest statement in the “Confidential to Editor” section, and submit your "Accept" recommendation.

Reviewer #1: All comments have been addressed

Reviewer #3: All comments have been addressed

2. Is the manuscript technically sound, and do the data support the conclusions?

Reviewer #1: Yes

Reviewer #3: Yes

3. Has the statistical analysis been performed appropriately and rigorously? 

Reviewer #1: Yes

Reviewer #3: Yes

4. Have the authors made all data underlying the findings in their manuscript fully available?

Reviewer #1: Yes

Reviewer #3: Yes

5. Is the manuscript presented in an intelligible fashion and written in standard English?

Reviewer #1: Yes

Reviewer #3: Yes

6. Review Comments to the Author

Reviewer #1: Authors have addressed all the queries and concerns. The Article may be accepted in the revised form

Reviewer #3: The authors have address the reviewer's comments. The manuscript has been revised asper the suggestions.

7. PLOS authors have the option to publish the peer review history of their article (what does this mean?). If published, this will include your full peer review and any attached files.

Reviewer #1: No

Reviewer #3: **Yes: **NAMRATA ANAND

---

## [Editor Report · Acceptance letter]

2 Apr 2024

PONE-D-24-00531R1 

PLOS ONE

Dear Dr. Asrat, 

I'm pleased to inform you that your manuscript has been deemed suitable for publication in PLOS ONE. Congratulations! Your manuscript is now being handed over to our production team.

Kind regards, 

on behalf of

Dr. Yash Gupta 

Academic Editor

PLOS ONE